# Features of Self-Diffusion of Tridecane Molecules in a Porous Medium of Kaolinite Used as a Model of a Chemically Inert Membrane

**DOI:** 10.3390/membranes13020221

**Published:** 2023-02-10

**Authors:** Aleksander Maklakov, Nariman Dvoyashkin, Elena Khozina

**Affiliations:** 1Institute of Physics, Kazan Federal University, 420008 Kazan, Russia; 2Almetyevsk State Oil Institute, 423400 Almetyevsk, Russia; 3A.N. Frumkin Institute of Physical Chemistry and Electrochemistry of Russian Academy of Science, 119071 Moscow, Russia

**Keywords:** self-diffusion, PMFG NMR, spin echo, tridecane, porous material, kaolinite, membrane, temperature, vapor phase, fast molecular exchange, activation energy

## Abstract

The present work focused on the experimental study of the specific features of self-diffusion of tridecane molecules in macroporous kaolinite, which is used as a raw material for the production of chemically inert membranes. The measurements of self-diffusion coefficients by pulsed magnetic field gradient nuclear magnetic resonance (PMFG NMR) revealed an increased translational mobility of tridecane molecules in kaolinite with incomplete filling of the pore space. This effect was accompanied by a sharp change in the slope of the Arrhenius plot of the self-diffusion coefficients of tridecane molecules in kaolinite. An analysis of the diffusion spin echo decay in the tridecane–kaolinite system revealed a discrepancy between the experimental data and the theoretical predictions, considering the effect of the geometry of porous space on molecular mobility. It was shown that the experimental results could be interpreted in terms of a model of two phases of tridecane molecules in the pores of kaolinite, in the gaseous and adsorbed state, coexisting under the fast-exchange conditions. Within the framework of the model, the activation energies of self-diffusion were calculated, which agreed satisfactorily with the experimental data. Additionally, the effects of the internal magnetic field gradients arising in a porous medium loaded with a gas or liquid on the data of the PFG NMR measurements were calculated. It was shown that the effect of magnetic field inhomogeneities on the measured self-diffusion coefficients of tridecane in kaolinite is small and could be neglected.

## 1. Introduction

The widespread use of membranes in various branches of industry and science [1] implies the need to obtain actionable knowledge about the physical processes occurring when the substances transported through them are separated into components [2]. It is important to understand the mechanism of mass transfer, the nature of the interaction of the molecules of the transferred substance with each other and with the surface of the membrane pores, etc. Real membranes differ in their properties, which determine the efficiency of membrane transport, and as a result, their applications in gas separation [3,4], liquid biofuel production [5], radioactive waste management [6,7], oil-in-water separation [8], etc. In addition to such properties of membranes as mechanical strength, thermal and chemical resistance, thermal and electrical conductivity and heat capacity, the structural and morphological characteristics of a membrane material are also essential since they determine the specific features of molecular transport of the target substance in membranes and membrane sorption efficiency [2,9,10,11]. In turn, the diffusion and sorption coefficients determine an essential parameter of the efficiency of a membrane: its permeability. The values of membrane permeability for the target substance provide information on coupled adsorption and diffusion properties of a membrane material. Thus, to decouple these properties, it is necessary to measure independently diffusion and adsorption parameters in membranes [12]. A well-characterized chemically inert porous material (clays, sands, porous glasses, etc.) filled with a liquid can serve as a model system for investigating the features of diffusion properties in membranes.

Recently, kaolin clays were proposed as a cheap alternative raw material for producing ceramic membranes, which are used in oil-field-produced water treatment [13,14] or in the conversion of biomass-derived free fatty acids to infrastructure compatible hydrocarbons [15]. Samhari and coworkers managed to fabricate a flat ceramic microfiltration membrane from natural kaolinite for seawater pretreatment and wastewater purification [16].

Based on the foregoing, in the present study, we considered kaolinite as a membrane model and tridecane as a model substance transported through this membrane. It should be noted that tridecane, as one of the major petroleum hydrocarbons, is present in wastewaters and soil [17,18] and its removal requires different methods, including membrane technology.

To study the molecular mobility of tridecane in kaolinite (a model system simulating mass transfer in a membrane), pulsed magnetic field gradient nuclear magnetic resonance [19] was employed. PMFG NMR is one of the most informative experimental methods for studying molecular mobility in porous media [20,21,22,23,24,25,26].

One of the unusual features of molecular mobility of liquid in a porous medium is abnormally high self-diffusion coefficients (SDCs) observed under certain experimental conditions. Boss and Steiskal [27] were the first to notice that the measured values of the SDC of water in the pores of hydrated vermiculite exceeded the bulk value; they attributed this effect to a contribution from a vapor phase at the boundaries of the porous material. Later, Karger et al. [21,28] also observed the enhanced mobility of water in NaY-zeolite with mesoporous hierarchical structure. They interpreted it using an analog of the Einstein equation for diffusion by uncorrelated jumps, according to which the mean life times between succeeding jumps of molecules in porous space become shorter as the pore filling decreases. D’Orasio et al. found similar effects for water in mesoporous glasses [29]. Almost simultaneously and independently, we measured the molecular mobility of water and a series of saturated hydrocarbons in pores of natural clay minerals using the PFG NMR method [30] and also discovered the anomalous concentration dependences of SDC, when the measured SDC values of a liquid that partially filled the pore space can exceed the bulk value.

In a series of PMFG NMR experimental studies on self-diffusivities of polar (water) and nonpolar hydrocarbons in model porous systems such as silica glasses [31,32,33], the effects of pore size, pore filling and temperature on the observed phenomenon were revealed. The abnormal molecular transport behaviors of hydrocarbons were investigated experimentally in nanoporous solids with hierarchical pore structure for two cases of mass exchange between micro- and mesopores [34,35,36]. The different temperature-dependent patterns of the concentration dependence of molecular mobility in micro- and mesopores were interpreted, taking into account the models of adsorption [24,34,35,36]. Bukowski et al. analyzed various theoretical approaches and simulation models for describing molecular transport in various micro- and mesoporous materials (porous glasses, zeolites, metal–organic framework), including the phenomenon of enhanced diffusion, and the calculated data were comparted with the experimental results obtained by PMFG NMR and quasi-elastic neutron scattering [37].

As follows from numerous experimental data, the enhanced diffusion is a phenomenon that is specific to fluid molecules in nanoporous materials under conditions of incomplete filling of pore space [27,28,29,30,31,32,33,34,35,36,37]. The ranges of pore filling and temperatures at which this phenomenon is observed depend on the physicochemical properties of both a fluid and a porous material.

In the present work, we continue to study the physical nature of this phenomenon and consider an effect of temperature. We also check a possible impact of a systematic error, which is inherent in PFG NMR measurements of SDC of liquid in porous materials due to a difference between their magnetic susceptibilities.

## 2. Materials and Methods

Tridecane (C_13_H_28_), hereinafter referred to as TD, of a chemically pure grade was purchased from Chimmed Group. The physicochemical properties of TD are as follows: molecular mass *M* = 184.4 g/mol, liquid density *ρ_l_* = 0.742 g/cm^3^ (at *T* = *T*_cr_ = 674.2–676.2 K), boiling point temperature *T*_0_ = 508 K [38,39].

Kaolinite is a typical layered silicate mineral with a rigid structural cell; most of the kaolinite minerals are close to the ideal formula Al_2_Si_2_O_5_(OH)_4_. It is composed of thin pseudohexagonal sheets of triclinic crystal. Its structure consists of stacks of electrically neutral 1:1 layers with 0.72-nm spacings. The adjacent layers are held together by hydrogen bonds between the basal oxygen atoms of the tetrahedral sheet and the hydroxyl groups of the exterior plane of the octahedral sheet [40]. Kaolinite particles have a thickness ranging from 0.05 µm to 0.1 µm and a length ranging from 0.2–0.5 µm to 0.5–1.0 µm [40,41,42].

According to experimental reports [41,42,43], kaolinite adsorbs low-molecular substances, including hydrocarbons, but their adsorption is limited to the surface of the particles (plane or edges), while the interlayer space remains inaccessible.

The kaolinite specimen was received from the Institute of Geology and Petroleum Technologies of Kazan Federal University (Kazan, Russia). The porous structure of kaolinite is characterized by a specific surface area *s*_1_ of 9.9 m^2^/g and a specific pore volume *V_p_* of 0.36 cm^3^/g evaluated from the adsorption measurements [44,45]. The maximum thickness of the plates achieved 0.1 μm.

The bulk-to-bulk method of sample preparation for the PMFG NMR measurements was described in a previous work [45]. The mass content of TD in the sample was defined as follows:$$\omega_{1}=m_{1}/\left(m_{1}+m_{2}\right),$$
here, *m*_1_ is the mass of TD injected into a NMR tube loaded with a weighed portion of kaolinite with mass *m*_2_. The samples with *w*_1_ = 0.038, 0.06, 0.079, 0.14 and 0.40 were studied.

The degree of pore filling of kaolinite with TD was calculated:$$\theta =V_{l}/V_{p}=\omega_{1}/\left(1-\omega_{1}\right)\rho_{l}V_{p},$$
here, *V_l_* is the specific volume occupied by tridecane. The *θ* values for all the samples were less than 1, except for that with *w*_1_ = 0.40 (see Table 1).

Translational mobility characteristics of TD were evaluated from the diffusional spin echo decay (DD) *A*(g^2^) recorded on a self-designed NMR pulse spectrometer (Department of Molecular Physics of Kazan Federal University) operating at a proton resonance frequency of 64 MHz. The spectrometer is equipped with a PMFG unit, providing a maximum gradient magnitude of 40 T/m.

The diffusion experiments used a standard stimulated echo sequence that includes three radio frequency (RF) pulses and two pulses of magnetic field gradients [19]. The stimulated echo sequence is shown schematically in Figure 1.

The observation time of diffusion is defined as:$$t_{d}=\left(\Delta -\frac{1}{3}\;\delta \right),$$

Here, δ is the duration of field gradient pulses, which did not exceed 2.2 ms; Δ is a time interval between the field gradient pulses. The diffusion time varied within the range of 3–15 ms.

In the case of isotropic free diffusion of a pure substance, the diffusion decay (DD) of the stimulated echo is described by an exponential function:(1)$$A\left(g^{2}\right)=A\left(0\right)exp\left(-\gamma^{2}\;\delta^{2}g^{2}Dt_{d}\right),$$
here, γ is the gyromagnetic ratio of proton.

The diffusion measurements in heterogeneous systems often yield spin echo decays that are complex in shape and cannot be fitted by an exponential function. In this case, an effective or average self-diffusion coefficient $\bar{D}=D_{eff}$ is introduced as a quantitative measure of translational molecular mobility throughout the sample under study. Its value is determined from the initial slope ($g^{2}\to 0$) of diffusion decay $A\left(g^{2}\right)$.

In our experiments, all PFG NMR parameters were calibrated using a standard liquid, namely, distilled water, which has a SDC value of 2.7 ± 0.1 10^−9^ m^2^⁄s at *T* = 303 K.

## 3. Results and Discussion

Figure 2 shows the diffusional spin echo decays *A*(*g*^2^) for sample 2 (*w*_1_ = 0.06) at different temperatures. One can see that at *T* = 303 K, the experimental DD curve (open circles) has a shape close to exponential, i.e., it can be approximately described by Equation (1), which is valid for isotropic free diffusion.

With an increase in temperature up to 383 K, the DD noticeably deviates from the exponential function (solid circles). A similar trend in the change in the DD shape with increasing temperature (and, as a consequence, with increasing values of measured SDC) was found theoretically for molecules in an infinitely slit-like pore with a width a [46].

According to [46], in this case, the DD is expressed as follows:(2)$$\frac{A\left(g^{2}\right)\;}{A\left(o\right)\;}=exp\left(-Dk^{2}t_{d}\right)\cdot \int_{0}^{1}exp\left\{k^{2}\left[Dt_{d}-\frac{\langle r_{⊥}^{2}\rangle }{2}\right]x^{2}\right\}dx.$$

Here, $\frac{\langle r_{⊥}^{2}\rangle }{2}=\frac{a^{2}}{3}\cdot \left[1-exp\left(\frac{3Dt_{d}}{a^{2}}\right)\right]$, and $\langle r_{⊥}^{2}\rangle$ is the square root-mean-square displacement of a molecule in a direction perpendicular to the pore walls. The DD curves calculated by Equation (2) with a = 0.05 μm are shown by solid lines in Figure 2.

Figure 3 shows the concentration dependences of effective SDC $\bar{D}$ determined from the initial slope of DD at different temperatures. It can be seen that at low temperatures of 294, 312, and 333 K, the measured SDC value of TD becomes smaller, as its content in kaolinite decreases (curves 1–3). This result is quantitatively consistent with the theoretical predictions [47,48,49]. However, an increase in temperature from 294 to 333 K is accompanied by a noticeable weakening of the dependence $\bar{D}\left(\omega_{1}\right)$. Moreover, at *T* ≥ 358 K, the character of the $\bar{D}\left(\omega_{1}\right)$ dependence changes dramatically (curves 4 and 5 in Figure 3). As the concentration of TD in kaolinite decreases, the observed SDC starts to increase up to a value exceeding the SDC of bulk liquid TD (*D_L_*) under the same conditions, i.e., $\bar{D}-D_{L}>0$.

At elevated temperatures, the effect of enhanced diffusion of TD is manifested at a larger filling of pores compared to that for lower temperatures (compare curves 4 and 5 in Figure 3). Thus, in the region of enhanced diffusion, parameter $\left(\bar{D}-D_{L}\right)$ increases both with a rise in temperature *T* at *w*_1_ = const and with a decrease in the pore filling at *T* = const. It should be emphasized that an increase in the measured $\bar{D}$ is observed only for the cases of partial filling of kaolinite pores with TD, i.e., for θ < 1.

We considered in more detail the influence of temperature on the effect of enhanced diffusion. For this purpose, we examine the temperature dependences of the effective value of $\bar{D}$ of TD for three samples represented in Figure 4 as an Arrhenius plot: $\left(lg\bar{D}-1/T\right)$. It can be seen that both for sample 4 with *w*_1_ = 0.40 or *θ* ≥ 1 (curve 3) and pure tridecane (curve 4), the dependence $\mathrm{lg}\bar{D}=f\left(1/T\right)$ is described by a straight line. However, for all other samples with incompletely filled pores (curves 1 and 2), the plots can be approximated by two straight lines with different slopes. Each of these lines is characterized by the apparent activation energy of diffusion of TD in the pores of kaolinite: Ei=−Rlge·ΔlgD¯Δ1T,
here, *R* is the gas constant; subscript *i* = *l* or *h* specifies the low- and high-temperature regions, respectively; *E_l_* and *E_h_* are the activation energies of self-diffusion of TD in kaolinite under the low- and high-temperature conditions, respectively.

Table 1 lists the values of *E_l_* and *E_h_* for all the samples. As follows from Figure 4, the changes in the slope of $lg\bar{D}\left(1/T\right)$ plots occur within a temperature range of 320–330 K that coincides with the temperatures at which the dependence $\bar{D}\left(\omega_{1}\right)$ becomes less strong. It can be seen from Table 1 that the values of $E_{l}$ calculated for samples 1–3 differ from the activation energy of diffusion for bulk TD, and moreover, it depends on the TD content. These facts can be attributed to the adsorption of TD molecules onto the surface of kaolinite. We assume that TD molecules are uniformly distributed over the pore surface. According to the data reported in [50], the volume of a TD molecule is 235·10^−30^ m^3^, and the corresponding area is 42 × 10^−20^ m^2^. From knowing the total number of TD molecules in the sample and the specific surface area of kaolinite, it was possible to calculate the number of monolayers formed by physically adsorbed TD molecules *n*_mol_ on the pore surface for each system TD–kaolinite. The values of *n*_mol_ are given in Table 1.

Following to [51], the activation energy of TD can be represented as
$$E_{l}=\alpha E_{ads}$$

Here, *E*_*ads*_ is the adsorption energy or the energy of van der Waals interactions between TD molecules and adsorption sites of kaolinite, the parameter α (0 < α ≤ 1) is determined by the TD–kaolinite van der Waals interactions. In these ranges of pore fillings and temperatures, the activation energy *E_l_* describes the random jumps of TD molecules from one adsorption site to another. It should be noted that with an increase in the number of TD monolayers on the kaolinite surface, the apparent activation energy corresponds to two processes: (1) stochastic hopping and (2) formation of vacancies for TD molecules diffusing within the surface layer [37]. Therefore, in this case, α > 1. In terms of this concept, one can interpret the differences in the values of *E_l_* for sample 1 and samples 2 and 3. However, a precise picture of TD diffusion within the surface layer requires a detailed knowledge of the surface structure of kaolinite in order to evaluate the TD–kaolinite interactions.

As mentioned above, the Arrhenius plot $lg\bar{D}\left(1/T\right)$ with the constant slope (and constant energy of activation) was observed for sample 4 (*θ* > 1), which is indicative of the fact that the translational mobility of TD molecules in the completely filled kaolinite pores is determined by the same mechanism over the entire range of studied temperatures. Latour et al. [52] showed that at long diffusion observation times (which is valid for our experiments), the diffusion constant of molecules in pores depends on the connectivity (tortuosity) of the porous space. Moreover, the long-time diffusion constant of a fluid in the membranes depends on their permeability and arrangement [11,52]. It should be noted that a similar Arrhenius plot was found for oversaturated mesoporous Vycor glass with n-pentane at the temperatures exceeding the boiling point [53]. Next, we discuss the diffusion behaviors of TD in kaolinite pores at elevated temperatures. As follows from Table 1, in the case of incomplete filling of pores with TD, the values of activation energy *E_h_* are sufficiently high. Moreover, the values of *E_h_* calculated for samples 1–3 exceed the molar heat of vaporization of TD: *q*_0_ = 45.67 kJ/mol at the boiling point $T_{0}\;$= 508–509 K [42].

It seems reasonable to assume that at elevated temperatures, in the samples with incomplete filling of kaolinite macropores (*θ* < 1), TD molecules exist (1) in the adsorbed state in the surface layers with properties close to those of liquid TD (except the first layer) and (2) in the state of saturated vapor. We suppose that these phases coexist under conditions of a fast exchange in the time scale of diffusion observation by the PMFG NMR technique. Namely, the fast exchange condition is realized if the lifetimes of TD molecules in the surface layer and in the gaseous state, *t_L_* and *t_G_*, respectively, are much less than diffusion observation time *t_d_*:(3)$$t_{L},\;t_{G}\ll t_{d},$$

We estimate the values of $t_{L}$ and $t_{G}$ at a temperature of 333 K, which is higher than that at which the changes are observed in the slope of the $lg\bar{D}\left(1/T\right)$ plot. Assuming again the spherical shape of TD molecule, we calculate its diameter: ~8 × 10^−10^ m. Therefore, the total thickness of the surface layer of adsorbed TD molecules *l* can be calculated as *l = n*_mol_ × 8 × 10^−10^ m. Hence, the time during which the molecule covers a distance equal to *l* can be calculated: $t_{L}^{\prime }\sim \frac{l^{2}}{2D_{L}}=$ 3 × 10^−8^ s. However, not every molecule can overcome the surface tension forces, reach the layer–gas interfacial boundary, and finally, pass into the gas phase. Thus, as a rough approximation, the real lifetime of TD molecules in the surface layer was calculated as follows: $t_{L}\approx t_{L}^{\prime }\cdot \frac{\rho_{L}}{\rho_{G}}$, where $\rho_{L}$ and $\rho_{G}$ are the TD liquid and saturated vapor phase densities, respectively. In the studied temperature range, the relation $\frac{\rho_{L}}{\rho_{G}}$ is approximately 10^−2^ [39] and $t_{L}$ ~3 × 10^−6^ s. The calculated value of *t_L_* is much shorter than the diffusion observation time $t_{d}$ ~3 × 10^−3^ s.

It should be noted that similar diffusion behaviors were found for nonpolar cyclohexane and water in Vitrapor#5 porous glass with partially filled micrometer pores, which were also attributed to a vapor phase contribution [32,33]. Valiullin et al. also interpreted the experimental data on self-diffusion of hexane in a model nanoporous glass with micrometer pores by molecular exchange between the liquid and vapor phases and evaluated the lifetimes of hexane molecules in the liquid phase from the analysis of the dependence of the DD shape on the diffusion observation time [31]. It should be noted that the effect of enhanced diffusion was not found for tridecane partially filling the interparticle space in quartz sand (macroporous system with a specific surface of ~10^−2^ m^2^/g) [54] and for benzene in microporous activated carbons with a specific surface of ~10^3^ m^2^/s) [24]. Thus, a pore size and specific surface area are the crucial factors that determine the relevance of molecular exchange limits relative to the diffusion observation time in the PMFG NMR experiments.

For TD in kaolinite pores, the lifetime in the vapor phase $t_{G}$ can be taken as the time during which the molecules overcome a distance equal to the pore size, which we consider comparable with the maximum thickness of the kaolinite plates *h* ~10^−7^ m. If we take into account that the SDC for tridecane in the vapor phase $D_{G}$ is of order 10^−5^ m^2^⁄s [39]), then, we obtain $t_{G}$ ~$\frac{h^{2}}{2D_{G}}$~5 × 10^−10^ s, which is also much less than $t_{d}$, and condition (3) is satisfied in the time scale of the PMFG NMR diffusion measurements.

In this case, the measured average value SDC for TD can be represented as a sum:(4)$$\bar{D}=N_{G}D_{G}+N_{L}D_{L},$$

Here, *N_G_* and *N_L_* are the relative number of TD molecules in the vapor and liquid phases, respectively.

Valiullin and coauthors in a series of works [34,35,36] developed a generalized model for the self-diffusion of low-molecular liquids in nanoporous solids based on the gas–liquid interphase exchange determined by the adsorption isotherm. They found that the effective SDC of liquid in nanopores reaches a maximum at concentrations corresponding to one-two monolayers. Below this concentration, the effective self-diffusivity is determined both by an increasing fraction of molecules in a vapor phase and an increasing surface self-diffusivity or *D_L_*. Above this concentration, a decrease in the effective SDC values with increasing concentration is a result of a decrease in the effective volume of the vapor phase with a minimum influence of the surface interactions and insignificant changes in SDC of molecules in the surface multilayers. A contribution of possible capillary condensation in mesopores to the molecular mobility exhibits an almost constant value of effective SDC.

As follows from Table 1, the smallest pore filling of kaolinite with TD corresponds to an approximately 6 monolayer coverage of kaolinite. Hence, the effective self-diffusion of TD in the pores of kaolinite is determined by the density and diffusivity of molecules in the vapor phase. Keeping in mind that $N_{L}=1-N_{G}$, and $D_{G}\gg D_{L}$, we can simplify Equation (4):(5)$$\bar{D}-D_{L}=N_{G}D_{G},$$

Then, from Equation (5), it follows that the temperature dependence $\left(\bar{D}-D_{L}\right)=f\left(T\right)$ is determined by the dependences $D_{G}\left(T\right)$ and $N_{G}\left(T\right)$.

In general, the self-diffusion of gas molecules in nanopores is described by an interpolation formula known as Bosanquet’s approximation [55,56], combining the molecular bulk (*D_b_*) and Knudsen (*D_K_*) diffusion, which are determined by the gas–gas and gas–pore wall collisions:$$\frac{1}{D_{G}}=\frac{1}{D_{b}}+\frac{1}{D_{K}}$$

We assume that with a concentration range under study, the amount of TD molecules in the vapor phase in the pore space formed by interconnected macropores is sufficient to ensure the predominance of the gas–gas collisions. Therefore, the contribution of Knudsen diffusivity becomes insignificant. This assumption seems reasonable, especially when taking into account the values of apparent energy *E_h_* exceeding the molar heat of TD vaporization. According to the molecular-kinetic theory of gases, the temperature dependence of SDC is described by the well-known formula:(6)$$D_{G}\left(T\right)=b\;T^{\frac{3}{2}},$$
where b is a constant.

To determine the dependence $N_{G}\left(T\right)$, we assume that saturated steam partially filling the pores obeys the ideal gas laws. It can be also surmised that the degree of pore filling can be expressed as follows:$$\theta =V_{L}/(V_{L}+V_{G}),$$
and the fraction of molecules in pores in the vapor phase:$$N_{G}=\frac{n_{G}V_{G}}{n_{G}V_{G}+n_{L}V_{L}},$$
here, $V_{L}$ and $V_{G}$ are volumes occupied by TD molecules in the liquid and vapor phases, $\;n_{L}$ and $\;n_{G}$ are the number of TD molecules per unit volume in the liquid and vapor states, respectively.

Then, at θ ≥ 0.05, which is true for all the studied samples, we can evaluate the relative number of TD molecules in the vapor phase using a formula:(7)$$N_{G}\approx \frac{n_{G}}{n_{L}}\cdot \frac{1-\theta }{\theta }=\frac{P}{n_{L}k_{B}T}\cdot \frac{1-\theta }{\theta },$$
here, P is the saturated vapor pressure of TD at temperature *T*; $\;k_{B}$ is the Boltzmann constant.

The temperature dependence of saturated vapor pressure is determined by the Clausius–Clapeyron equation:$$\;\frac{dP}{dT}=\frac{q\left(T\right)}{T\left(V_{\mu }{}^{G}-V_{\mu }{}^{L}\right)},$$
where q is the molar heat of vaporization, $V_{\mu }{}^{G}$ and $V_{\mu }{}^{L}$ are the molar volumes of saturated vapor and liquid, respectively.

Usually, $V_{\mu }{}^{G}\gg V_{\mu }{}^{L}$, and, therefore, the last equation can be written as follows:(8)$$\frac{dP}{P}≅\frac{q\left(T\right)dT}{RT^{2}}.$$

It is known that the heat of vaporization *q* is a temperature-dependent parameter:$$q\left(T\right)=\left(U_{\mu }{}^{G}-U_{\mu }{}^{L}\right)+P_{G}\left(V_{\mu }{}^{G}-V_{\mu }{}^{L}\right)\approx \left(U_{\mu }{}^{G}-U_{\mu }{}^{L}\right)+RT.$$

Here, the difference $\left(U_{\mu }{}^{G}-U_{\mu }{}^{L}\right)$ is a change in the internal energy of one mole of a substance during a liquid–vapor transition at any temperature, $P_{G}\left(V_{\mu }{}^{G}-V_{\mu }{}^{L}\right)$ is the value of the system expansion work.

At the boiling point $T_{0}$ for TD, we obtain:$$q\left(T_{0}\right)=q_{0}\approx \left(U_{0\mu }{}^{G}-U_{0\mu }{}^{L}\right)+RT_{0},$$

Here, $q_{0}$ is the specific heat of vaporization (or boiling) at the boiling point $T_{0}$, and the difference $\left(U_{0\mu }{}^{G}-U_{0\mu }{}^{L}\right)$ is a change in the internal energy of one mole of a substance during a liquid–vapor transition at the boiling point $T_{0}$.

Keeping in mind that:$$U_{\mu }{}^{G}=U_{0\mu }{}^{G}-\overset{T_{0}}{\underset{T}{\int }}c_{v}{}^{G}\left(T\right)dT=U_{0\mu }{}^{G}-c_{v}{}^{G}\left(T_{0}-T\right)$$

And:$$U_{\mu }{}^{L}=U_{0\mu }{}^{L}-\overset{T_{0}}{\underset{T}{\int }}c^{L}\left(T\right)dT=U_{0\mu }{}^{L}-c^{L}\left(T_{0}-T\right),$$
where $c^{L}$ and $c_{v}{}^{G}$ are the molar heat capacity of the liquid and gas at constant volume, respectively; $U_{0\mu }{}^{G}$ and $U_{0\mu }{}^{L}$ are the molar values of the internal energies of the gas and liquid at the boiling point $T_{0}$, respectively.

It should be noted that the molar heat capacities are constant in the temperature range of 300–400 K; hence, we can write:(9)$$q\left(T\right)=q_{0}+\left(c^{L}-c_{P}{}^{G}\right)\cdot (T_{0}-T).$$

Here, $c_{P}{}^{G}=c_{v}{}^{G}+R$.

By substituting Equation (9) for Equation (8) and integrating the resulting expression, we obtain:(10)$$lnP=-\frac{q_{o}+\left(c^{L}-c_{p}{}^{G}\right)T_{0}}{R}\cdot \frac{1}{T}-\frac{c^{L}-c_{p}{}^{G}}{R}lnT+A,$$

Here, A is the integration constant.

Then, the substitution of Equations (6), (7), and (10) into Equation (5) leads to the expression:(11)$$ln\left(\;\bar{D}-D_{L}\right)=-\frac{q_{0}+\left(c^{L}-c_{p}{}^{G}\right)T_{0}}{R}\cdot \frac{1}{T}+\left(\frac{1}{2}-\frac{c^{L}-c_{p}{}^{G}}{R}\right)lnT+const,$$

Notably, the second term in Equation (11) depends on temperature much weaker than the first term. In this case, the dependence of $ln\left(\;\bar{D}-D_{L}\right)$ on 1/*T* must be described by a linear function.

Next, we introduce the following notations:(12)$$q_{0}+\left(c^{L}-c_{p}{}^{G}\right)T_{0}=E_{h}{}^{T}$$
and
$$\left(\frac{1}{2}-\frac{c_{v}{}^{L}-c_{p}{}^{G}}{R}\right)lnT+const=B.$$

Then, Equation (11) can be rewritten as follows:(13)$$ln\left(\bar{D}-D_{L}\right)=-\frac{E_{h}{}^{T}}{R}\cdot \frac{1}{T}+B.$$

It follows from Equation (13) that the dependence of $ln\left(\bar{D}-D_{L}\right)$ on 1/T*n* can be fitted by a linear function, the slope of which is defined as $q_{0}+\left(c^{L}-c_{p}{}^{G}\right)T_{0}=E_{h}{}^{T}$ and its magnitude is independent on the amount of a fluid in the pore space.

In Figure 5, we compared the experimental data on the TD self-diffusivity for samples 1 and 3 with different amounts of TD plotted as a function $\mathrm{lg}\left(\bar{D}-D_{L}\right)=f(1/T)\;$ for temperatures above 333 K (see Figure 5).

The values of $D_{L}$ are the results of extrapolating the curves $lg\bar{D}(1/T)$ with slope $E_{l}/R$ (Figure 4). Two conclusions can be drawn from the results presented in Figure 5. First, the experimental dependences $lg\left(\bar{D}-D_{L}\right)\;\mathrm{versus}\;(1/T)\;$ are linear, and second, their slopes, which determine an apparent activation energy for both samples ($E{\;}_{h}^{’}$ = 74 kJ/mol (1) and $E{\;}_{h}^{’}$ = 68 kJ/mol (2)), rather weakly depend on the content of TD in kaolinite. This result directly follows from Equation (11). The values of apparent activation energy $E{\;}_{h}^{’}$ are also given in Table 1.

Next, we estimate the calculated parameter $E_{h}{}^{T}$ from Equation (12) and compare it with the experimental found value $E{\;}_{h}^{\prime }$ (see Equation (13)). Using the reference data for TD $T_{0}$= 509 K, $q_{0}$ = 45.67 kJ/mol, $c_{p}{}^{G}$ = 0.304 kJ/mol, $c^{L}$ = 0.385 kJ/mol [39,57] and Equation (12), we evaluated the value of $E_{h}{}^{T}$ ~87 kJ/mol.

Therefore, it can be seen that the calculated and experimental values $E_{h}{}^{T}$ and $E{\;}_{h}^{\prime }$ are close (see Table 1).

Relationship (10) made it possible to calculate the pressures of TD-saturated vapors over the range of the studied temperatures, which are consistent with the values reported in the literature [57]. Using Equations (5) and (7) and the values of $D_{G}$ [39], $D_{L},$ and $N_{G}$ obtained from the extrapolation of the low-temperature section of the $lg\bar{D}(1/T)$ plot (see Figure 4), we calculated the high-temperature branches of $lg\bar{D}(1/T)$ plots for samples 1 and 3. The calculated dependences $1^{\prime }\;\mathrm{and}\;2^{\prime }$ shown in Figure 4 satisfactorily describe the experimental data.

In principle, one can expect a change in the slope of the $lg\bar{D}(1/T)$ plots since, as follows from Equation (6), with increasing temperature, the amount of TD molecules in the vapor phase should be enough to define the temperature dependence of $\bar{D}$. The preliminary experiments on the diffusivity of decane inserted in kaolinite confirmed our proposal. Moreover, according to the experimental data reported in [53], the similar increase in the slope of the Arrhenius plot for *n*-pentane completely filling the mesopores in Vycor glass, which was observed at the temperatures above the boiling point, could be attributed to a contribution of the vapor phase in a pore space free of liquid. Finally, we considered the effect of random magnetic fields created by a difference in the magnetic susceptibilities of kaolinite and tridecane on the self-diffusion parameters measured by the PMFG NMR method in order to evaluate a systematic error in the measurements. Fatkullin derived analytical expressions for the SDC measured under conditions of the random magnetic field created by the difference in the magnetic susceptibilities of the components of a spatially inhomogeneous medium [58]. He showed that the corrections to the measured SDC for this effect depend on the ratio of three characteristic times: the correlation time $\tau_{c}$ of a molecule moving in a random magnetic field B* with a correlation radius ξ: $\tau_{c}=\xi^{2}/D$; the time interval $\tau_{1}$ between the first and second RF pulses in the stimulated echo sequence (see Figure 1), and the diffusion observation time $t_{d}$. Depending on the ratio between these times, three regimes are distinguished:

long correlation time $\tau_{c}$≫$t_{d}$≫$\tau_{1}$;intermediate time $t_{d}$≫$\tau_{c}$≫$\tau_{1}$;short correlation times $t_{d}$≫$\tau_{1}$≫$\tau_{c}$.

In most of our measurements, we used $t_{d}=6\;\mathrm{ms}$; $\tau_{1}=1.5\;\mathrm{ms}$. It seems impossible to accurately calculate the correlation time $\tau_{c}$ since there are no direct methods for determining the correlation radius of a random magnetic field $B^{*}$. Nevertheless, it seems reasonable to assume that this radius is related to the linear size of the pores. The slit-like pores in kaolinite are comparable with the thickness of hexagonal plates. Therefore, with radius ξ ~0.05 μm and D ~10^−9^ m^2^/s (the value determined before the enhanced diffusion, see Figure 4), we obtain: $\tau_{c}\;$~2 × 10^−6^ s. Therefore, the short correlation time regime $t_{d}$≫$\tau_{1}$≫$\tau_{c}$ is implemented in our experiments. As a result, the apparent SDC measured under the condition of random magnetic fields [58] is determined as follows:(14)$${\bar{D}}^{*}=\bar{D}\left(1+\frac{4}{9\pi^{3/2}}\cdot \frac{\gamma^{2}\langle B^{*2}\rangle \tau_{1}{}^{3/2}\xi^{2}a_{0}}{{\bar{D}}_{0}{}^{3/2}\cdot \;t_{d}}\right),$$
where $\bar{D}$ is the SDC measured in the absence of random fields; $a_{0}$ is the minimum linear size of the system (in our case, it is the thickness of the liquid layer on the pore surface: $a_{0}$ ~10^−8^ m); $\langle B^{*2}\rangle$ is the average square of the magnetic field induction expressed by a formula:(15)〈B*2〉12 ≈4πχp−χLH0,
here, $\chi_{p}\;\mathrm{and}\;\chi_{L}$ are the bulk magnetic susceptibilities of the porous medium and liquid, respectively.

By using the Gouy method [59], we obtained: $\chi_{p}$ = + 0.52∙10^−6^ (g^−1^) and $\chi_{L}$ = −0.65∙10^−6^ (g^−1^). In our experiments, the magnetic field was $H_{0}$ = 14 kG. Then, we evaluated the second term in brackets in Equation (15) and found that the value of correction to ${\bar{D}}^{*}$ of 10^−4^ is negligible. Thus, the effect of random magnetic fields did not lead to the distortion of the measured values of effective SDC of TD in the pores of kaolinite. This conclusion was supported by the results of similar studies of enhanced self-diffusion of other hydrocarbon liquids, including natural oil in kaolinite [45,54].

## 4. Conclusions

The PMFG NMR measurements of diffusion behaviors of TD filled in kaolinite revealed the enhanced translational mobility in the case of partial filling of pores, which became more pronounced with temperature. This effect, as well as the specific features of the temperature dependence of the effective self-diffusion measurements, were interpreted in terms of a two-phase model under conditions of fast exchange between the liquid and gas phases of the diffusant in the partially filled pores. The contribution from the TD molecules in the state of saturated vapor determined the deviation of the $lg\bar{D}(1/T$) plots from the linear Arrhenius function. The activation energies of self-diffusion of TD in the partially filled pores of kaolinite were evaluated to characterize the confined translational mobility of TD in the surface layer molecules at low temperatures, which resulted from the fast exchange with the vapor phase of TD, the contribution from which increases with temperature. The results of calculations based on the two phase exchange model were in agreement with the experimental data.

It was shown that the effect of magnetic field inhomogeneities had no effect on the measured SDC values of TD in pores of kaolinite and, therefore, could not be the cause of the observed specific features of diffusion behaviors.

These findings provide an insight into the mechanism of molecular transport through membrane materials and can be used for their designing and applications under different thermodynamic conditions.

## Figures and Tables

**Figure 1 membranes-13-00221-f001:**
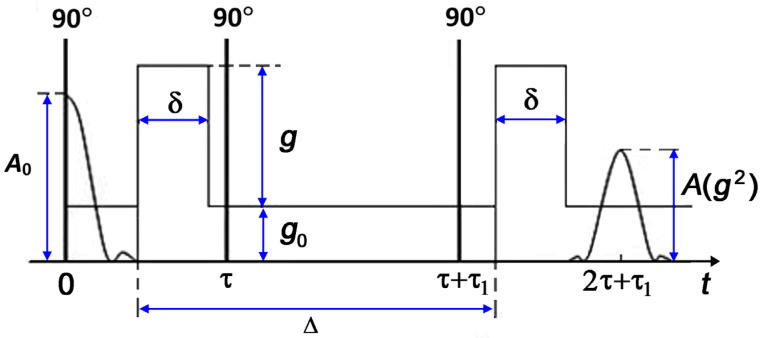
Three-pulse sequence with two pulses of field gradient for measuring SDC from the diffusional decay of stimulated echo [19]: 90° are RF pulses; $A\left(g^{2}\right)$ and $A_{0}$ are the spin echo amplitudes in the presence and absence of magnetic field gradient pulses with intensity g and duration δ, respectively; Δ is the time between magnetic field gradient pulses; $g_{0}$—is constant gradient of magnetic field; τ is the interval between the first and second RF pulses; $\tau_{1}$ is the interval between the second and third RF pulses.

**Figure 2 membranes-13-00221-f002:**
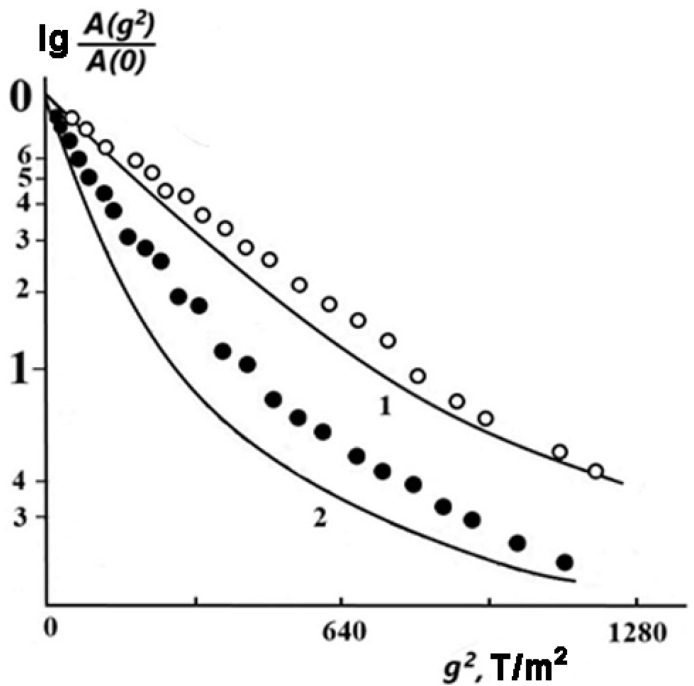
Diffusion decays of spin echo in sample 2 ($\omega_{1}$ = 0.06) at *T* = 303 K (open circles) and at *T* = 383 K (solid circles). Diffusion time $t_{d}$ is 3 ms and duration of PMFG δ is 0.2 ms. Solid curves were calculated by Equation (2) [46] for *T* = 303 K (curve 1) and 383 K (curve 2); a = 0.05 µm.

**Figure 3 membranes-13-00221-f003:**
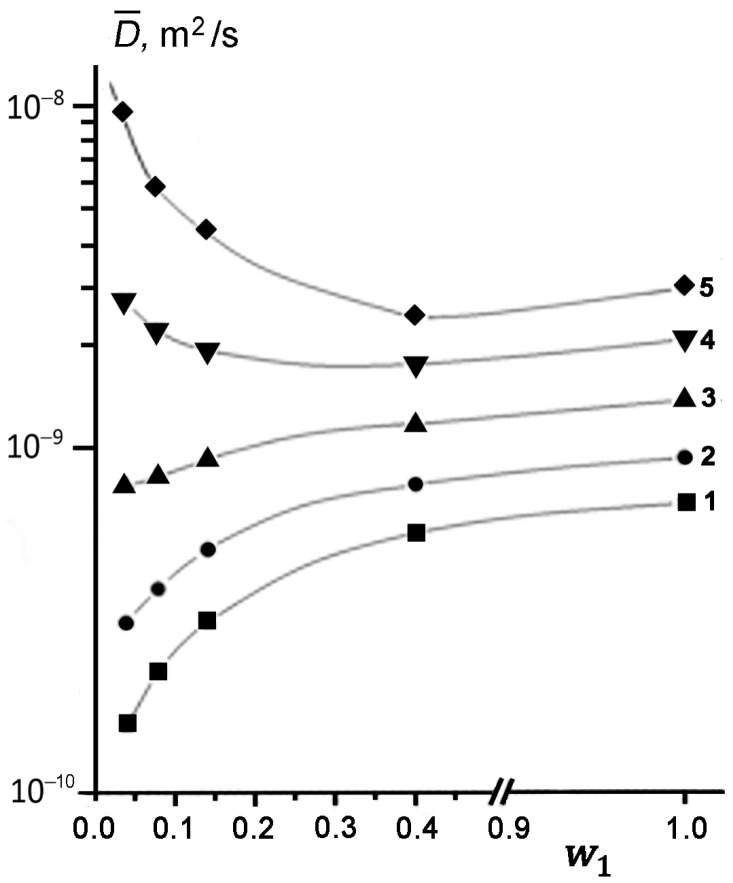
Concentration dependences of the effective (average) SDC of TD in kaolinite measured at a temperature, K: 294 (1), 312 (2), 333 (3), 358 (4), and 384 (5). Symbols are the experimental data, and solid lines are the results of spline-approximation.

**Figure 4 membranes-13-00221-f004:**
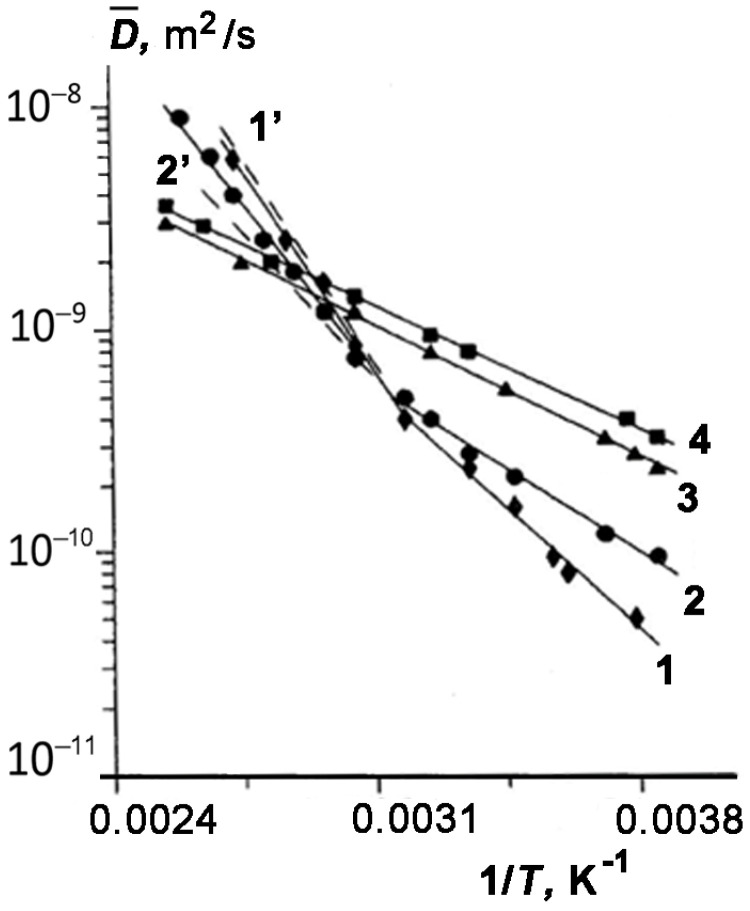
Temperature dependences of effective SDC of TD enclosed in pores of kaolinite at different values of $\omega_{1}$: 0.038 (1), 0.079 (2), 0.40 (3) and 1 (4), i.e., for samples 1, 3, 4 and 5, respectively. The dashed lines (1′ and 2′) show the results of calculations according to Equation (5) with consideration of Equations (7) and (10) for samples 1 and 3.

**Figure 5 membranes-13-00221-f005:**
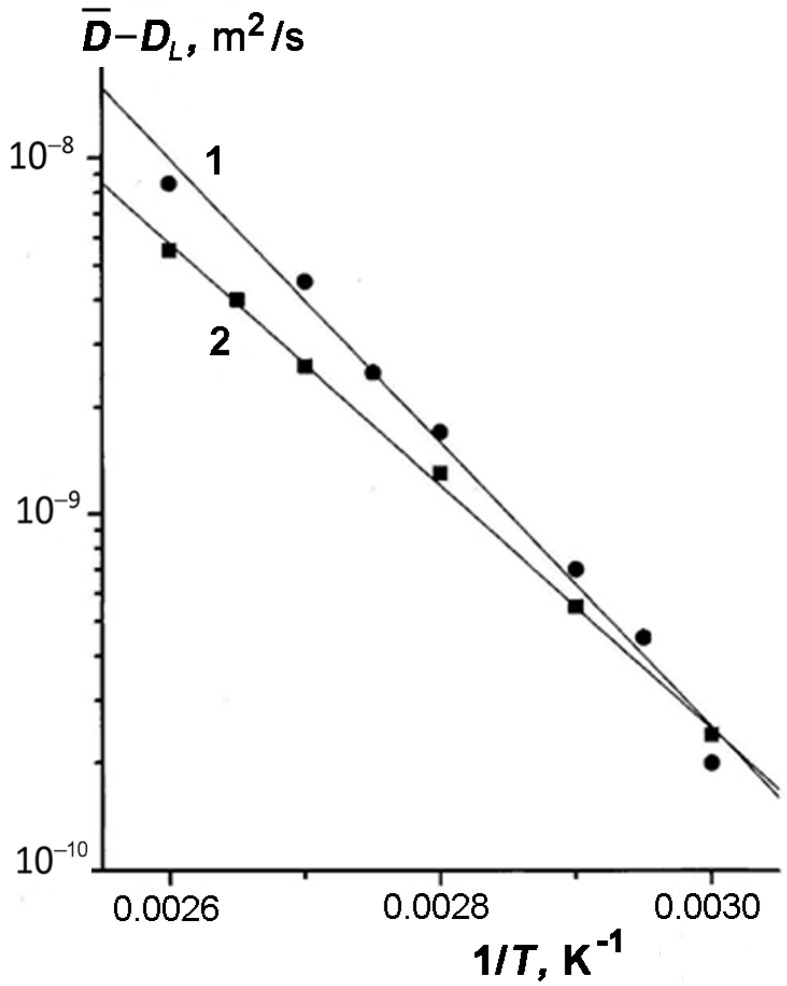
Temperature dependences $\left(\bar{D}-D_{L}\right)$ of TD enclosed in pores of kaolinite for samples 1 with $\omega_{1}$ = 0.038 (1) and 3 with $\omega_{1}$ = 0.079 (2). Symbols show the experimental data; solid lines are the linear approximation.

**Table 1 membranes-13-00221-t001:** Characteristics of the samples, including the relative amount of TD in kaolinite, degrees of pore filling, number of TD monolayers, energies of activation of TD diffusivity in pores of kaolinite.

No	$\omega_{1}$	θ	$n{\;}_{mol}\;$	$E_{l},\;\mathrm{kJ}/\mathrm{mol}$	$E_{h},\;\mathrm{kJ}/\mathrm{mol}$	$E_{h}^{\prime },\;\mathrm{kJ}/\mathrm{mol}$	$E_{h}^{T},\;\mathrm{kJ}/\mathrm{mol}$
1	0.038	0.15	~6	28.8	50.6	74	87
2	0.060	0.24	~10	24.3	48.2	-	87
3	0.079	0.33	~13	19.8	46.0	68	87
4	0.400	>1	-	16.8	-		-
5	1.000	-	-	15.5	-		-

Here, $n{\;}_{mol}\;$ is the number of TD monolayers on the pore surface of kaolinite (calculated below); ***E_l_***, $E_{h}$, $E_{h}^{\prime }$, and $E_{h}^{T}$ are the activation energies of TD self-diffusion in kaolinite, which are evaluated from the temperature dependence of SDC (explanations are given below in the text).

## Data Availability

Not applicable.

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
