# Peer review of "Features of Self-Diffusion of Tridecane Molecules in a Porous Medium of Kaolinite Used as a Model of a Chemically Inert Membrane"

_membranes, 2023, doi:10.3390/membranes13020221_

Round 1

Reviewer 1 Report

 Line 36: What is the difference between structural and morphological characteristics?

Lines 41, 55, 69, 135, 154, 164, 259, 263: the term “introduced” is not the best choice. I suppose, “enclosed” will be better for this case.

Line 61: “the physical nature” repeated twice.

Equation (1): Please specify what is the multiplier dD at the end of the equation.

Lines 94, 97: Please explain the notation “RF”.

Line 125: “On” should be deleted.

Lines 136, 147, 155, 157, 164, 267, 268, 269. What means “cr.1”?

 Style of writing should be corrected: sometimes T, another case T.

 English should be corrected.  

Author Response

We express our sincere gratitude to you for your comments and suggestions on the presented article, which, of course, allowed us to improve the text of the article.

Line 36: In our understanding, structure is understood as the mutual arrangement of the constituent parts of an object at the microscopic level (for example: "The structure of the crystal lattice of an object"). By morphological features, we understand the structural features of the constituent parts of an object at the macroscopic level (for example: "the lamellar structure of a crystalline object or its shape in the form of spherulites").

Lines: 41, 55, 69, 135, 154, 164, 259, 263: In all cases, the term "introduced" is replaced by the term "enclosed".

Equation (1): Multiplier "dD" removed.

Line 61: The extra phrase "the physical nature" has been removed.

Lines 94, 97: "RF" stands for "radio frequency".

Line 125: "On" - removed.

Lines: 136, 147, 155, 157, 164, 267, 268, 269: in all cases, "cr.” replaced by "curve".

The style of writing "T" is reduced to a single "T" everywhere.

Reviewer 2 Report

In the present work by Dvoyashkin et al. self-diffusion of tridecane in a model chemically inert membrane, i.e. natural clay mineral kaoloinite, has been studied by means of PFG NMR. An enhanced molecular mobility has been observed upon lowering the liquid content in the porous matrix. The enhanced mobility was associated with the fast molecular exchange between tridecane and its saturated vapor. Altogether, this is an interesting study which suits the scopes of the journal. After revisions suggested below, including also English proofread, the manuscript might be recommended for publication.

Revisions recommended:

1)    The phenomenon of enhanced molecular diffusion in partially saturated porous solids is known since late 1980s. Since the pioneering experiments this phenomenon has been thoroughly studied and well-documented in the literature. In this regard, the reference list needs to be accordingly updated. In the present form, it is far from being representing the current state of the art in this research field.

2)    Caption of Table 1 needs to be expanded and all quantities explained.

3)    In Eq.1 remove “dD” 

4)    Eq.1 is followed by “where k = …”, however Eq.1 does not contain “k” 

5)    Figure 4 contains inappropriate axis labels, the same with Figure 3

6)    Line 180: “The values of Ein all samples with < 1 (table 1) are 180 quite high and exceed the values of the molar heat of vaporization TD equal to 181 ?= 45.67 kJ/mol determined at the boiling point” What is experimental error in the determination of Eh to justify this statement. I would rather say that the experimentally obtained Eh are very close to the heat of vaporization.

7)    Line 223 with the following up Eq.6 poses a critical problem. It is assumed that diffusion in the vapor phase is captured by a molecular diffusion. The T^3/2 dependency is typical of bulk gaseous phases. However, in the samples studied, the gaseous phase is found between the kaolinite plates with the dimension of about 50 nm as stated by the authors (see, e.g., Figure 2). In this case, diffusion follows the patterns of Knudsen diffusion with a weaker temperature dependency T^1/2 only, where the mean free path is temperature independent and is given by the distance between the kaolinite plates. In turn, this questions all subsequent discussion and its reliability.  Excellent reproduction of the activation energies for diffusion in the course of the liquid-gas phase transitions by using the concept of the Knudsen diffusion has already been reported in the literature. 

Author Response

We appreciate the time and effort that you dedicated to providing feedback on our manuscript and are grateful for the insightful comments.  Your inputs are very helpful for improving the manuscript. We have revised our manuscript accordingly and included almost all your suggestions and clarified the text when needed. Detailed responses are given in the attached file.

Round 2

Reviewer 2 Report

The authors have revised their manuscript in response to the reviewers' comments. I am partially satisfied with the revisions. The present work still does not cover adequately the current state. In particular, I missing references to a series of comprehensive studies of the same phenomena and respective discussions. In particular,

the diffusion enhancement has been thoroughly discussed in:

J Chem Phys, 120 (2004) 11804; Langmuir, 28 (2012) 3621; Magn Reson Imaging, 23 (2005) 285

the activation energies during phase coexistences are discussed in:

Phys Rev E, 75 (2007) 041202; Microporous and Mesoporous Materials, 164 (2012) 273; Microporous and Mesoporous Materials, 178 (2013) 84

and many other publications.

Author Response

We are grateful for the insightful comments. Your inputs are very helpful for improving the manuscript. We have revised our manuscript accordingly and included almost all references you suggested to the text. Detailed response is given in the attached file.
